# Phage Endolysins as Potential Antimicrobials against Multidrug Resistant *Vibrio alginolyticus* and *Vibrio parahaemolyticus*: Current Status of Research and Challenges Ahead

**DOI:** 10.3390/microorganisms7030084

**Published:** 2019-03-18

**Authors:** Nandita Matamp, Sarita G. Bhat

**Affiliations:** Department of Biotechnology, Cochin University of Science and Technology, Kochi-682022, Kerala, India; nandita.matamp@gmail.com

**Keywords:** antibiotic resistance, endolysin, *Vibrio alginolyticus*, *V. parahaemolyticus*

## Abstract

*Vibrio alginolyticus* and *V. parahaemolyticus,* the causative agents of Vibriosis in marine vertebrates and invertebrates, are also responsible for fatal illnesses such as gastroenteritis, septicemia, and necrotizing fasciitis in humans via the ingestion of contaminated seafood. Aquaculture farmers often rely on extensive prophylactic use of antibiotics in farmed fish to mitigate *Vibrios* and their biofilms. This has been postulated as being of serious concern in the escalation of antibiotic resistant *Vibrios*. For this reason, alternative strategies to combat aquaculture pathogens are in high demand. Bacteriophage-derived lytic enzymes and proteins are of interest to the scientific community as promising tools with which to diminish our dependency on antibiotics. Lysqdvp001 is the best-characterized endolysin with lytic activity against multiple species of *Vibrios*. Various homologues of *Vibrio* phage endolysins have also been studied for their antibacterial potential. These novel endolysins are the major focus of this mini review.

## 1. Introduction

*Vibrio alginolyticus* and *V. parahaemolyticus*, the normal inhabitants of estuarine and marine environments, are notable human enteropathogens associated with seafood-borne mortality and illness worldwide. *V. alginolyticus* is reported as the etiological agent of wound and ear infections (both otitis media and otitis externa), intracranial infection, peritonitis and osteomyelitis among many others [1,2,3], while *V. parahaemolyticus* causes bacterial gastroenteritis associated with the consumption of raw or undercooked seafood [4,5].

In aquaculture, the hazard of infectious diseases has led to significant stock losses and problems with animal welfare. Hence, intensive aquaculture promotes the indiscriminate use of anti-microbials, thereby causing the dissemination of antimicrobial-resistant (AMR) bacteria and resistance genes in aquaculture products and the environment [6]. This global concern has necessitated the exploration of alternative therapies for bacterial pathogens in animal production, especially in aquaculture. Amongst the several substitutes, that include probiotics, essential oils and anti-microbial peptides, phage therapy has gained much attention for preventing and controlling pathogenic infections in aquaculture facilities. Recent advances in phage genome sequencing have kindled the application of phage encoded enzymes, especially endolysins, as biocontrol and therapeutic agents against major food-borne pathogens.

There have been numerous reviews on endolysins as antimicrobials against Gram-positive bacteria. However, in this review, we concentrate on phage lysin biology against Gram-negative pathogens *V. alginolyticus* and *V. parahaemolyticus*. The endolysin characteristics that are important to combatting multidrug resistant *Vibrios* are summarized, thereby outlining the remarkable potency of these enzymes in the mitigation of similar pathogens in aquaculture.

## 2. Antibiotic Resistance in *Vibrios alginolyticus* and *V. parahaemolyticus*

Antimicrobial/chemotherapeutic agents against *V. alginolyticus* and *V. parahaemolyticus* are used either as feed additives and/or as immersion bath solutions in fish farms. The recommended antibiotics against *Vibrios* are fluoroquinolones (ciprofloxacin, levofloxacin), tetracyclines (doxycycline, tetracycline), third-generation cephalosporins (cefotaxime, ceftazidime, ceftriaxone), aminoglycosides (amikacin, apramycin, gentamicin, streptomycin) and folate pathway inhibitors (trimethoprim-sulfamethoxazole) [7]. The excessive use of antibiotics has led to the evolution of numerous strains that exhibit resistance to a single or a combination of antibiotics. However, as reported by the United States Centers for Disease Control and Prevention (CDC), the occurrence of *Vibrio*-related infections has increased dramatically since 2001 [8] (CDC, 2016).

Gram-negative bacteria have developed divergent mechanisms to bypass the inhibitory effects of antibiotics such as (1) drug inactivation/destruction (2) decreased antibiotic penetration and efflux (3) target site modification and (4) global cell adaptations [9,10,11]. Genes and associated insertion elements, which confer antibiotic resistance, are usually found localized in plasmids as multi-resistance regions (MRR) in these organisms [12]. Studies of antibiotic resistance in many pathogens such as *V. cholera, Staphylococcus aureus* and *Salmonella* have been reported, but the mechanism of the same in *V. alginolyticus* and *V. parahaemolyticus* is poorly documented [13].

Fluoroquinolones, a promising class of broad-spectrum antibiotics, are direct inhibitors of DNA synthesis. They bind to the enzyme-DNA complex and stabilize DNA strand breaks created by the enzymes DNA gyrase and topoisomerase IV. [14]. Kitaoka and his co-workers have reported the presence of spontaneous chromosomal mutations in gyrA and parC genes in *Vibrios* that encode subunits of DNA gyrase and topoisomerase IV, respectively. These mutations could alter the affinity of theses enzymes, thus protecting *Vibrios* from quinolones. [15]. Self-transmissible plasmids that confer resistance by plasmid-mediated quinolone resistance (PMQR) mechanisms were also being investigated in *V. parahaemolyticus* [16]. Tetracycline has been recommended as the antimicrobial of choice for the treatment of severe *Vibrio* infections, thanks to its ability to inhibit the synthesis of pathogenic extracellular enzymes [17]. The emergence of *V. alginolyticus* possessing tet plasmids for tetracycline resistance [18] and pVAS3-1 plasmids for β-lactamase resistance is alarming [19].

### Treatment Costs for Vibriosis

According to CDC Outbreak Surveillance Data, around 6680 cases of *V. parahaemolyticus* and 165 cases of *V. alginolyticus* have been reported annually. The annual health costs of *Vibrio* infections are estimated to be over $30 million. These data are quite imprecise due to limitations of surveillance data and underreporting. Under such circumstances, the costs are likely higher, leading to considerable uncertainty into the overall estimate. In addition to the hike in treatment costs, antimicrobial resistance can lead to protracted hospital stays and escalations in morbidity and mortality rates [20].

## 3. Bacteriophage Endolysins-‘the Holy Grail’ to Control Food Borne Pathogens

Bacteriophages or phages are viruses that specifically infect and lyse bacteria. Following their discovery by Twort and Felix D’Herelle, it became clear that they exhibit two kinds of life cycles: lytic (used by both virulent and temperate phages) and lysogenic (used exclusively by temperate or pro-phages) [21]. Lytic or virulent phages have evolved a lytic system to weaken the bacterial cell wall, leading to bacterial lysis. This bacterial lysis is achieved by phage-encoded muralytic enzymes called Endolysins (or lysins) that degrade the peptidoglycan (PG) layer present in the bacterial cell wall during the final stage of the phage reproduction events. The lysis events of double-stranded DNA bacteriophages can be elucidated by three different mechanisms. The most explicitly demonstrated mechanism is canonical lysis, where lysins act on PG layer with the help of a second phage encoded protein called holin, in a timely-controlled fashion [22]. Holins depolarize the cytoplasmic membrane by allowing endolysins to diffuse through pores in the membrane and target the PG layer. The second pathway is mediated by a special class of holins designated as pinholins which forms small, heptametrical channels in the membrane instead of large holes as seen in canonical lytic pathway. These pinholins work in association with Signal-arrest-release (SAR) endolysins which are inactive tethered enzymes accumulated in the periplasm. Using proton motor force (PMF), pinholins trigger the activation of these pro-enzymes, refolding their configuration leading to their release from the bi-layer, thereby degrading PG. Pinholins act as timers for endolysin activation playing no lead role in their export. In Gram-negative hosts, the lysis of OM is by a third functional class of lysis proteins called the spanins [23]. Spanin complex consists of small outer membrane lipoprotein (o-spanin) and an integral cytoplasmic membrane protein (i-spanin) which disrupts OM by 3 modes: (i) enzymatic degradation of PG cross links [24] (ii) pore formation [25] and (iii) inner membrane-outer membrane fusion [26]. Phage researchers have termed these enzymes as ‘enzybiotics’; they can be exploited for their ability to kill variety of pathogens [27].

## 4. Gram-Negative Endolysins as Antimicrobials

### Basic Structure and Function

The peptidoglycan layer is the major structural component of the bacterial cell wall responsible for protection, physical integrity and shape. It is composed of chains of alternating residues of N-acetylmuramic acid (MurNAc) and N-acetylglucosamine (GlcNAc), connected by β-1,4 glycosidic bonds, linked to a short stem of tetrapeptide [28]. The cell wall of Gram-negative organisms has an outer membrane (OM) situated above a thin PG layer and the limited permeability of OM [29] poses a major hurdle for development of novel antimicrobials against Gram-negative pathogens preventing many compounds from reaching their intracellular targets. Since the endolysin susceptible layer (PG) is found between an inner and outer membrane, effective strategies, like use of peptides, detergents, and chelators, should be applied in combination with hydrolytic enzymes to improve the applicability of phage lysins. As an example, 5 mM EDTA used in combination with *E. coli* phage endolysin PlyE146 400 μg/mL, decreased titers of E. coli K12 by ca. 2 log_10_ CFU/mL upon 2 hours of incubation. [30]. Moreover, the peptide moiety made of L- and D-amino acids is highly conserved (chemotype A1γ) in Gram-negative organisms, whereas the carbohydrate backbone is conserved in both Gram-positive and negative bacteria.

Phage endolysins are analogous in structure and function to bacterial lysins, and are closely related to the small family of mammalian PG recognition proteins [31]. They can have either a globular or modular structure. Endolysins from phages infecting Gram-negative hosts are mostly small single-enzymatically active domain (EAD) globular proteins (molecular mass 15–20 kDa) without a specific cell wall binding domain (CBD) module [32,33]. An EAD cleaves a specific bond in the PG structure, whereas a CBD targets the EAD to its substrate by binding PG or another cell wall component. Apart from these two domains, recent reports of some Gram-negative antibacterial endolysins have revealed another domain CHAP (cysteine,histidine-dependent amidohydrolase/peptidase) belonging to amidase family whose role is to facilitate hydrolysis of the PG layer [34,35]. This feature enables them to enhance their catalytic skills by binding to multiple sites during cell lysis. An endolysin isolated from a phage infecting a Gram-negative species is therefore enzymatically-active on the PG of any other Gram-negative strain [36].

The first endolysins infecting Gram-negative bacteria were reported in the 1960s, and were mostly encoded by T-phages infecting *Escherichia coli*. Earlier, they were simply referred to as ‘lysozymes’ based on their functional similarity to egg white lysozyme, a muralytic enzyme well noted for its anti-bacterial activity. Later, based on the mechanism of action, PG hydrolases were classified into 4 groups: (a) glycosidases which cleave the glycan component of peptidoglycan, (b) amidohydrolase, that cleaves amide bond between the glycan moiety (MurNAc) and the peptide moiety (l-alanine) of the PG (c) endopeptidase which cleaves peptide bonds between two amino acids, and finally, (d) lytic transglycosylases that cleave the β(1-4) linkages between NAM and NAG residues of the PG. Transglycosylases are not true hydrolases, as they do not require water to catalyze PG cleavage. Most of the endolysins reported so far are lytic transglycosylases. The complexity of endolysins can be further illustrated by the fact that an elaborate motif search of approximately 723 putative endolysins in database has revealed the presence of 24 types of catalytic domains, 13 binding domains, and 89 possible architectural organizations [37].

The modular structure of endolysin has facilitated development of engineered lysins with desired properties such as higher stability, solubility and broad killing spectrum. Because of the independent functions of N-terminal catalytic domain (CD) and a C-terminal cell-wall binding domain (CBD), lysins can be constructed by fusing them from different origins or with other molecules [38]. Among the engineered lysins, chimeolysins and artilysins are worth mentioning. Several chimeolysins have been constructed with extended broad spectrum activity against Gram-positive pathogens like *Staphylococcus, Streptococci* and *E. faecalis* [39,40,41,42,43]. Recently, novel chimeolysin (ClyF) active against planktonic and biofilm MRSA designed from a chimeolysin library with different combinations of CDs and CBDs was expressed in *E. coli* [44]. Artilysins are outer membrane-penetrating lysins constructed by fusing a fragment of natural lysin with peptides or other proteins with high anti-bacterial activity against Gram-negative pathogens. The lipopolysaccharide destabilizing peptides of artilysins can be effectively exploited against *Pseudomonas*, *E. coli*, *Salmonella* and *Yersinia* [45,46]. The concept of endolysin delivery against Gram-negatives is further expanded by the development of Innolysins which are constructed by combining receptor binding proteins (RBPs) of candidate phages. Zampara and his co-workers constructed twelve Innolysins using phage T5 endolysin and receptor binding protein Pb5, which bind irreversibly to the phage receptor FhuA involved in ferrichrome transport in *E coli*. It was proved that they pass through the outer membrane and degrade the PG layer, thereby killing the target bacteria [47].

## 5. *Vibrio* Phage Endolysins

Phage therapy experiments have shown promising results in the eradication of several pathogenic *Vibrios* (*V. harveyi*, *V. parahaemolyticus*, *V. alginolyticus*, *V. splendidus*, *V. anguillarum*,) in aquaculture settings since 1999 [48,49,50,51,52]. The extensive amount of genetic information assembled from phage whole genome sequencing has opened up new horizons to design novel antimicrobial agents. In this respect, timely exploration into utilization of *Vibrio* phage endolysins has sparkled interest among active phage researchers. Table 1 shows the complete list (to date) of all endolysins/putative ORFs coded by *V. alginolyticus* and *V. parahaemolyticus*. The three dimensional structures of *Vibrio* phage endolysins have been predicted by homology modeling (Figure 1).

### 5.1. Lysqdvp001 and Its Homologues

#### 5.1.1. Structure, Function and Physiochemical Properties

The endolysin Lysqdvp001 is derived from *V. parahaemolyticus* bacteriophage qdvp001, a lytic broad-spectrum phage belonging to *Myoviridae* family with genome length of 134,742-bp. [59]. The endolysin gene (ORF 60) of qdvp001 has a close relationship to *Vibrio* (*cholerae*) phage ICP1, which shares the same modular structure with ORF 60. The PG_binding _1 domain of the *Vibrio* phage ICP1 endolysin gene is 58 % homologous to ORF 60, whereas the CHAP domain shares a 66 % amino acid sequence identity. Interestingly, SMART analysis of endolysin has shown an unusual structure with two domains: a PG binding (PF01471) domain and a CHAP (PF05257) domain. Lysins with dual domains have been mostly reported in phages infecting Gram-positive pathogens and rare among bacteriophages infecting Gram-negative bacteria [65]. Lysqdvp001 is a modular endolysin with no transmembrane regions or signal peptide regions. Bioinformatic analysis also revealed the absence of any holins to assist the function of Lysqdvp001. The estimated molecular weight is 25.9 kDa and pI value is 5.97. The endolysin was cloned in competent *E. coli* BL21 Star^TM^ (DE3) strain and the recombinant endolysin has a good yield of 10.4 g from 300 mL of *E.coli* culture. Turbidity reduction assay of the purified product demonstrated promising results as the endolysin reduced turbidity of host bacteria by 0.6 log upon 5 min of incubation. This effective reduction was observed due to pretreatment of bacterial culture with EDTA for 5 min. Furthermore, Lysqdvp001 was able to lyse 11/11 *V. parahaemolyticus* strains tested, whereas the parent bacteriophage qdvp001 had a shorter host range of lysing 3/11 strains suggesting a broader anti-bacterial spectrum of the purified phage enzyme.

#### 5.1.2. LysVPMS1

LysVPMS1 was obtained from *V. parahaemolyticus* bacteriophage VPMS1. The host used to propagate the phage was isolated from shrimp farms in northwestern Mexico during an acute hepatopancreatic necrosis disease (AHPND) outbreak in 2014. This endolysin is the first reported phage lytic enzyme against *V. parahaemolyticus* AHPND strains. LysVPMS1 showed lytic activity against 17 AHPND strains and 5 non- AHPND strains. The highest rate of muralytic activity was observed in case of *V. parahaemolyticus* ATCC-17802 strain (96%). This information is quite significant in terms of the ability of purified LysVPMS1 to lyse strains from different origins with different degrees of pathogenicity. Moreover, this endolysin has the unique ability to lyse the cell wall of other *Vibrio* species specifically *V. alginolyticus*, *V. harveyi* and *V. campbellii*. More information on biochemical and bactericidal properties of the LysVPMS1 endolysin is presently unavailable [62].

#### 5.1.3. LysVPp1

VPp1 is a double-stranded DNA phage capable of infecting *V. parahaemolyticus* strains belonging to *Myoviridae* family. Its genome consists of 50,431 bp with a G+C content of 41.35%. The ability of VPp1 to reduce bacterial load during depurination in oysters was reported back in 2014 [36]. Recently, endolysin (LysVPp1) derived from VPp1 was purified and assessed for its anti-bacterial activities. LysVPp1 is a soluble lytic transglycosylase related to hen egg white lysozyme with a molecular weight of ~44 kDa and yield of 1 mg/mL. Peptidoglycan binding domain was not reported in LysVPp1.No holins/antiholin were also annotated in the phage genome. The antibacterial spectrum of the lysin was evaluated via two methods-(1) gel diffusion assay and (2) turbidity reduction assay. In gel diffusion assay, *V. parahaemolyticus* ATCC 17802 was used as the substrate. The hydrolase activity was determined by color changes around the holes in gels (0.01% potassium hydroxide + 0.1% methylene blue) containing peptidoglycan. Gel holes treated with LysVPp1 showed a light blue color resulting from the hydrolysis of peptidoglycan, thereby validating the hydrolytic activity of recombinant enzyme. In addition, the turbidity of EDTA-pretreated *V. parahaemolyticus* cells was reduced by 0.4 log after 5 min of incubation. The lytic spectrum assay of parent strain VPp1 lysed only *V. parahaemolyticus* isolates whereas the recombinant lysin LysVPp1 could hydrolyze 9 of 12 *Vibrio* strains tested, which included closely related *Vibrio* strains such as *V. parahaemolyticus*, *V. campbellii*, and *V. azureus* [61].

### 5.2. cwlQ- First Recombinant Endolysin with Holin Assistance

Vp670 is lytic phage of *Podoviridae* family capable of infecting *V. alginolyticus* strains. This is the first report of a *V. alginolyticus* phage whose lysis cassette was annotated, cloned and expressed. The genome size of Vp670 is 43,121 bp which codes for 49 ORFs and contains a lysis module, composed of two components- *holA* (holin) and *cwlQ* (endolysin). *cwlQ* is a relatively small protein(15–20 kDa) belonging to hydrolase-2 domain superfamily (Pfam 07486). TM pred analysis showed *holA* has a transmembrane helix with a hydrophilic C-terminal region inside the cytoplasmic membrane. Clones were expressed in *E. coli* (LPN028 and LPN030) and *V. alginolyticus* (LPN041 and LPN043) strains and these cells were able to survive under L-arabinose induction conditions. The clone expression was further studied by TEM analysis. Clones with *holA* and *cwlQ* had their OM layer disrupted and their cellular contents released from channels in the cell membrane. Cells without expressed genes had intact cellular structures with no morphological difference. Coexpression of both genes has resulted in severe cell damage compared to the expression of *holA* alone in the cells [57].

The therapeutic potential of the above reported endolysins been investigated in neither in vitro nor in vivo models.

## 6. Challenges of Endolysin Engineering and Delivery

There are numerous reports supporting the antibacterial activity of endolysins in vivo, but only a few of them have been proven by human clinical trials. There are numerous reports supporting the antibacterial activity of endolysins in vivo, but little information has been published on human clinical trials. SAL200 is the first endolysin based therapeutic formulation with a recombinant form of phage endolysin SAL-1 (rSAL-1) derived from the bacteriophage SAP-1, as its active pharmaceutical ingredient. SAP-1 infects *Staphylococci*, including MRSA and vancomycin-resistant *S. aureus* (VRSA) strains [66,67]. The first in-human phase 1 study of SAL200 provided preliminary information on safety, tolerability, pharmacokinetics, and pharmacodynamics of the product upon intravenous injection among healthy adults [68]. No serious adverse effects were observed in volunteers except mild and temporarily observed effects such as fatigue, headaches and myalgia. Similarly Staphefekt SA.100, a recombinant phage endolysin formulated ointment against infections caused by MRSA strains is available in a cetomacrogol-based cream/gel as over-the-counter treatment in Europe since 2017 [69].

Many challenges need to be addressed and overcome to deliver engineered chimeric endolysins. Lysins are non-replicating proteinaeous molecules with short half-life in systemic circulation [21,70]. They also elicit immunological response when applied systemically leading to catalytic loss of the enzyme [65]. Endolysins can be used in combination with other anti-bacterials, as they have been proven to act synergistically with antibiotics [71]. Currently, studies on lysin dosage are underdeveloped. Safe and successful therapeutic application of endolysin requires detailed information on bioavailability, immunogenicity and lysin synergy. More human clinical trials are anticipated to investigate phage endolysin treatments to combat several human pathogens. This is extremely important due to the substantial increase of multi-drug resistant pathogens and the steady decline in the discovery of new classes of antibiotics.

## 7. Conclusions

The extensive use of antibiotics has resulted in the emergence of multidrug resistant ‘superbugs’ worldwide. Bacteriophage encoded lytic enzymes ‘endolysins’ have enormous anti-microbial potential to fight against food borne pathogens in this multi-drug-resistance era. Promising results have encouraged active phage researchers to apply phage enzymes in various fields, such as food safety, pathogen detection, surface decontamination and nanotechnology. All reported endolysins showed a broad activity spectrum for the genus *Vibrio*. Research on *Vibrio* phage-encoded lytic enzymes has intensified since 2016. Lysqdvp001 and its homologues are highly divergent enzymes which are capable of superior lytic and antibacterial activity compared to their parent phages. Several attributes, such as high catalytic activity, modular structure and dual catalytic domains, support the robust development of them as novel alternatives to conventional antibiotic therapy. As the most abundant biological entity on earth, Bacteriophages’ lytic proteins are also considered structurally and functionally divergent. Bio-informatic and proteomic studies will allow researchers to expand endolysins as a powerful tool with diverse applications.

## Figures and Tables

**Figure 1 microorganisms-07-00084-f001:**
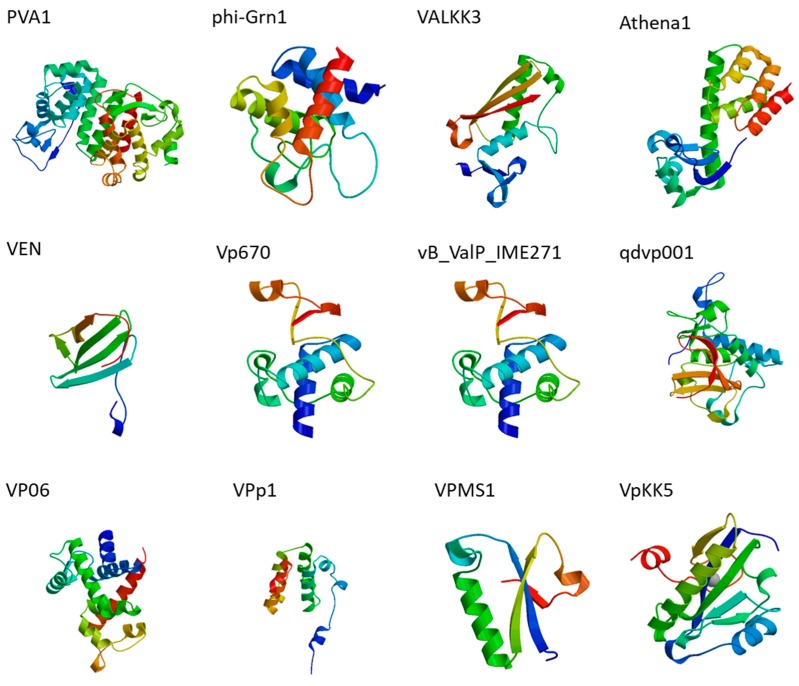
Structural modeling of *Vibrio* phage endolysins.

**Table 1 microorganisms-07-00084-t001:** Showing endolysins of *Vibrio alginolyticus* and *V. parahaemolyticus* phages.

Bacteria	Phage	Putative Endolysins/Predicted Orfs	Features	Reference
***Vibrio alginolyticus***	PVA1	gp60	Putative lysozyme family protein *	[50]
	phi-Grn1	phiGrn1_0012	SLT domain protein/endolysin *	[53]
	ValKK3	ORF304	Tail lysozyme	[54]
	Athena1	Cds006	Protein with lysozyme activity *	[55]
	VEN	gp50	Cell wall hydrolase-like protein *	[56]
	Vp670	cwlQ	endolysin	[57]
	vB_ValP_IME271	CDS64	endolysin	[58]
***V. parahaemolyticus***	qdvp001	Lysqdvp001	modular endolysin	[59]
	VP06	PP_00050	membrane-bound lytic murein transglycosylase *	[60]
	VPp1	LysVPp1	endolysin	[61]
	VPMS1	LysVPMS1	endolysin	[62]
	VpKK5	ORF62	*N*-acetylmuramoyl-l-alanine amidase *	[63]
	pTD1	BAW98403.1	Endolysin *	[64]

* indicates the newly identified features of sequenced endolysins as part of the present study.

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
