# Peer review of "Phage Endolysins as Potential Antimicrobials against Multidrug Resistant Vibrio alginolyticus and Vibrio parahaemolyticus: Current Status of Research and Challenges Ahead"

_microorganisms, 2019, doi:10.3390/microorganisms7030084_

Round 1

Reviewer 1 Report

This manuscript reviews the state of the art of vibrio bacteriophage lysins and their potential utility as antibacterial agents, which is at a very early stage (mostly in vitro activities demonstrated). It is generally well-written and well-researched.

 -There appears to be new data presented in Figure 1 showing predicted structures of most described/sequenced lysins -if it is not new, it needs to be referenced.

 -Figure 2 does not add much information to the text and can be removed.

-There are numerous minor grammatical issues:

-"gram-negative", " gram-positive", "entero-pathogens","multidrug-resistant" terms should be consistently hyphenated

-fluoroquinolone misspelled (line 49)

-line 60-change "An elaborated study" to "Studies..

-line 64-67 - no need to describe mechanisms of action of quinolones here

-line 80- change "inconclusive" to "imprecise" 

-line 80 " the costs could go substantially high" - should be "higher"

-line 102 -"cell wall .. have" ->should be "has"

-line 120-121 - add "the ->"belonging to [the] amidase family whose role is to facilitate hydrolysis of [the] PG layer"

-line 136 - "PG hydrolases are classified into 4 " add [groups]

line 137 - "glycosidases  which cleaves" -->should be "cleave"

line 149 -"till date" -should be " to date"

line 150 "are summarized here" can be deleted

section 5.1.1

reference 44 indicates the following observation worth noting: "the endolysin gene (ORF 60) of qdvp001 has a close relationship to Vibrio (cholera) phage ICP1, which shares the same modular structure with ORF 60. The PG_binding _1 domain of the Vibrio phage ICP1 endolysin gene is 58 % homologous to ORF 60, whereas the CHAP domain shares a 66 % amino acid sequence identity."

Clarify that the lysin gene was cloned in E coli - Suggest change to  "

The [cloned recombinant] endolysin had a good yield

 of 10.4 g from  300 mL of [E. coli] culture."

section 5.1.3

reference 46 includes the following result which should be noted- " In the lytic spectrum assay (Table 1), LysVPp1 could hydrolyze 9 of 12 Vibrio strains, which included V. parahaemolyticus, V. campbellii, and V. azureus"

-

Author Response

This manuscript reviews the state of the art of vibrio bacteriophage lysins and their potential utility as antibacterial agents, which is at a very early stage (mostly in vitro activities demonstrated). It is generally well-written and well-researched.

Dear Reviewer,

Thanks a lot for the valuable comments and we have incorporated all the modifications to the best of our knowledge.

 -There appears to be new data presented in Figure 1 showing predicted structures of most described/sequenced lysins -if it is not new, it needs to be referenced.

Swiss pdb modeling (Fig 1) of all sequenced lysins have been carried out as part of this review paper. Hence this is new.
In Table 1, the newly identified features (carried out as part of this review paper) of /sequenced lysins are indicated by star(*); all other information previously described are referenced .

 -Figure 2 does not add much information to the text and can be removed.

Figure 2 is removed in the revised manuscript

-There are numerous minor grammatical issues:

-"gram-negative", " gram-positive", "entero-pathogens","multidrug-resistant" terms should be consistently hyphenated

All corrections have been implemented in the revised manuscript

-fluoroquinolone misspelled (line 49)

Fluoroquinolone corrected in manuscript.

-line 60-change "An elaborated study" to "Studies..

Corrected

-line 64-67 - no need to describe mechanisms of action of quinolones here

Corrected

-line 80- change "inconclusive" to "imprecise"

Replaced inconclusive with imprecise

-line 80 " the costs could go substantially high" - should be "higher"

Corrected

-line 102 -"cell wall .. have" ->should be "has"

Corrected

-line 120-121 - add "the ->"belonging to [the] amidase family whose role is to facilitate hydrolysis of [the] PG layer"

Corrected

-line 136 - "PG hydrolases are classified into 4 " add [groups]

Corrected

line 137 - "glycosidases  which cleaves" -->should be "cleave"

Corrected

line 149 -"till date" -should be " to date"

Corrected

line 150 "are summarized here" can be deleted

Corrected

section 5.1.1

reference 44 indicates the following observation worth noting: "the endolysin gene (ORF 60) of qdvp001 has a close relationship to Vibrio (cholera) phage ICP1, which shares the same modular structure with ORF 60. The PG_binding _1 domain of the Vibrio phage ICP1 endolysin gene is 58 % homologous to ORF 60, whereas the CHAP domain shares a 66 % amino acid sequence identity."

Clarify that the lysin gene was cloned in E coli - Suggest change to "The [cloned recombinant] endolysin had a good yield of 10.4 g from  300 mL of [E. coli] culture."

The point worth mentioning was added in the text and E.coli (BL21 StarTM (DE3) with strain name has also been added.
The endolysin was cloned in competent E.coli BL21 StarTM (DE3) strain and the recombinant endolysin has a good yield of 10.4 g for from 300 mL of E.coli culture.

section 5.1.3

reference 46 includes the following result which should be noted- " In the lytic spectrum assay (Table 1), LysVPp1 could hydrolyze 9 of 12 Vibrio strains, which included V. parahaemolyticus, V. campbellii, and V. azureus"

Lytic spectrum of both parent phage strain and endolysin has been noted.
The lytic spectrum assay of parent strain VPp1 lysed only V. parahaemolyticus isolates whereas the recombinant lysin LysVPp1 could hydrolyze 9 of 12 Vibrio strains tested, which included closely related Vibrio strains such as V. parahaemolyticus, V. campbellii, and V. azureus.

Reviewer 2 Report

This minireview by Matamp and Bhat describes the research performed to date on Vivrio phage endolysins. Some parts are not well documented and some spell check is required.

Line 43: the species have already been named, so it should be V. alginolyticus and V. parahaemolyticus

Line 46: same. Names of species are italicized

Line 85: the holin-endolysins lysis is not an universal mechanism, it is only described in double-stranded DNA phages. On the other hand, lysis described in this section by the authors is the canonical lysis, there are also other mechanisms mediated by pinholins and spanins in Gram negatives. See R.Young, J Microbiol. 2014, 52(3): 243–25. Authors can explain this a bit more and not generalize.

Line 92: PG should be the complete word, since it has not been named so far

Line 95: here it should be only PG

Line 98: this section should be reorganized, it doesn't follow an structure for example: structure of PG, structure of endolysins, classification, mode of action and examples (including engineered endolysins).

Line 119: CHAP domain is an EAD and the role of all of them is the hydrolysis of the PG, so it is not special for that. CHAP domain seems to be in many gram positive endolysins, above all in S. aureus phages, and it is a very potent domain being the major responsible for the activity of the protein.

Line 124: before Innolysins, other engineered endolysins have been described againts Gram negative pathogens, the Artilysins, and nothing is said about them

Line 141: remove 'for'

Line 156: short name of the species

Line 157: phage families are italicized

Line 169: short name of the species

Line 174: short name of the species

Line 185: short name of the species

Line 186: phage families are italicized

Line 200: phage families are italicized

Line 206: species names are italicized

Line 208-211: gene names are italicized

Line 216: There already are in vivo clinical trials with endolysins. Several endolysins are in Phase 1 clinical trials and several companies are recruiting or have started Phase 2 clinical trials. This section must be rewritten.

There are a lot of spaces between words through all the text, and some phrases where a space is needed (see for example line 230), please check this.

Author Response

This minireview by Matamp and Bhat describes the research performed to date on Vivrio phage endolysins. Some parts are not well documented and some spell check is required.

Dear Reviewer,

Thanks a lot for the valuable comments and we have incorporated all the modifications to the best of our knowledge.

Line 43: the species have already been named, so it should be V. alginolyticus and V. parahaemolyticus

Corrected

Line 46: same. Names of species are italicized

Corrected

Line 85: the holin-endolysins lysis is not an universal mechanism, it is only described in double-stranded DNA phages. On the other hand, lysis described in this section by the authors is the canonical lysis, there are also other mechanisms mediated by pinholins and spanins in Gram negatives. See R.Young, J Microbiol. 2014, 52(3): 243–25. Authors can explain this a bit more and not generalize.

A note on three different lysis mechanisms by DNA phages have been added here.
The lysis events of double-stranded DNA bacteriophages can be elucidated by three different mechanisms. The most explicitly demonstrated mechanism is canonical lysis, where lysins act on PG layer with the help of a second phage encoded protein called holin, in a timely-controlled fashion [21]. Holins depolarize the cytoplasmic membrane by allowing endolysins to diffuse through pores in the membrane and target the PG layer. The second pathway is mediated by a special class of holins designated as pinholins which forms small, heptametrical channels in the membrane instead of large holes as seen in canonical lytic pathway. These pinholins work in association with Signal-arrest-release (SAR) endolysins which are inactive tethered enzymes accumulated in the periplasm. Using proton motor force (PMF), pinholins trigger the activation of these pro-enzymes, refolding their configuration leading to their release from the bi-layer thereby degrading PG. Pinholins act as timers for endolysin activation playing no lead role in their export. In Gram-negative hosts, the lysis of outer membrane is by a third functional class of lysis proteins called the spanins [22]. Spanin complex consists of small outer membrane lipoprotein (o-spanin) and an integral cytoplasmic membrane protein (i-spanin) which disrupts OM by 3 modes-(i) enzymatic degradation of PG cross links [23] (ii) pore formation [24] and (iii) inner membrane-outer membrane fusion [25]

Line 92: PG should be the complete word, since it has not been named so far

Corrected

Line 95: here it should be only PG

Corrected

Line 98: this section should be reorganized, it doesn't follow an structure for example: structure of PG, structure of endolysins, classification, mode of action and examples (including engineered endolysins).

The section is reorganized. Extra notes have been added on engineered lysins with examples.
The modular structure of endolysin has facilitated development of engineered lysins with desired properties such as higher stability, solubility and broad killing spectrum. Because of the independent functions of N-terminal catalytic domain (CD) and a C-terminal cell-wall binding domain (CBD), lysins can be constructed by fusing them from different origins or with other molecules [37]. Among the engineered lysins, chimeolysins and artilysins are worth mentioning. Several chimeolysins have been constructed with extended broad spectrum activity against Gram-positive pathogens like Staphylococcus, Streptococci and E.faecalis.[38-42]. Recently, novel chimeolysin (ClyF) active against planktonic and biofilm MRSA designed from a chimeolysin library with different combinations of CDs and CBDs was expressed in E. coli [43]. Artilysins are outer membrane-penetrating lysins constructed by fusing a fragment of natural lysin with peptides or other proteins with high anti-bacterial activity against Gram-negative pathogens. The lipopolysaccharide destabilizing peptides of artilysins can be effectively exploited against Pseudomonas, E.coli, Salmonella and Yersinia [44,45]. The concept of endolysin delivery against Gram-negatives is further expanded by the development of Innolysins which are constructed by combining receptor binding proteins (RBPs) of candidate phages. Zampara and his co-workers constructed twelve Innolysins using phage T5 endolysin and receptor binding protein Pb5, which bind irreversibly to the phage receptor FhuA involved in ferrichrome transport in E coli. It was proved that they pass through the outer membrane and degrade the PG layer thereby killing the target bacteria [46].

Line 119: CHAP domain is an EAD and the role of all of them is the hydrolysis of the PG, so it is not special for that. CHAP domain seems to be in many gram positive endolysins, above all in S. aureus phages, and it is a very potent domain being the major responsible for the activity of the protein.

Corrected

Line 124: before Innolysins, other engineered endolysins have been described againts Gram negative pathogens, the Artilysins, and nothing is said about them

Information on Artilysins with examples are added in the text. ( Item 6)

Line 141: remove 'for'

Corrected

Line 156: short name of the species

Corrected

Line 157: phage families are italicized

Corrected (Myoviridae)

Line 169: short name of the species

Corrected (V. parahaemolyticus)

Line 174: short name of the species

Corrected (V. parahaemolyticus)

Line 185: short name of the species

Corrected (V. parahaemolyticus)

Line 186: phage families are italicized

Corrected (Myoviridae)Line 200: phage families are italicized

Corrected (Podoviridae)

Line 206: species names are italicized

Corrected, E. coli (LPN028 and LPN030) and V. alginolyticus (LPN041 and LPN043)

Line 208-211: gene names are italicized

Corrected (holA and cwlQ)

Line 216: There already are in vivo clinical trials with endolysins. Several endolysins are in Phase 1 clinical trials and several companies are recruiting or have started Phase 2 clinical trials. This section must be rewritten.

Human clinical trials of two endolysins have been added and the section is modified.
There are numerous reports supporting the antibacterial activity of endolysins in vivo , but only little information on human clinical trials. SAL200 is the first endolysin based therapeutic formulation with a recombinant form of phage endolysin SAL-1 (rSAL-1) derived from the bacteriophage SAP-1, as its active pharmaceutical ingredient. SAP-1 infects staphylococci, including MRSA and vancomycin-resistant S. aureus (VRSA) strains [63,64].The first-in-human phase 1 study of SAL200 provided preliminary information on safety, tolerability, pharmacokinetics, and pharmacodynamics of the product upon intravenous injection among healthy adults [65]. No serious adverse effects were observed in volunteers except mild and temporarily observed effects such as fatigue, headaches and myalgia. Similarly Staphefekt SA.100, a recombinant phage endolysin formulated ointment against infections caused by MRSA strains is available in a cetomacrogol-based cream/gel as over-the-counter treatment in Europe since 2017 [66]. Many challenges need to be addressed and overcome to deliver engineered chimeric endolysins.

There are a lot of spaces between words through all the text, and some phrases where a space is needed (see for example line 230), please check this.

Corrected

Reviewer 3 Report

Lines 60-62:

The authors state that studies dealing with antibiotic resistance in V. alginolyticus and V. parahaemolyticus is poorly documented.  My question to this is why?  Is it that these pathogens, until recently, have not been clinically significant to warrant such investigations?  If not, then it would be good to explain some of the potential challenges in studying these Vibrio species which has resulted in inconclusive results, or no results in understanding antibiotic resistance in these 2 Vibrio species.

Author Response

Lines 60-62: The authors state that studies dealing with antibiotic resistance in V. alginolyticus and V. parahaemolyticus is poorly documented.  My question to this is why?  Is it that these pathogens, until recently, have not been clinically significant to warrant such investigations?  If not, then it would be good to explain some of the potential challenges in studying these Vibrio species which has resulted in inconclusive results, or no results in understanding antibiotic resistance in these 2 Vibrio species.

Dear Reviewer,

Thanks a lot for the valuable comments and we have incorporated all the modifications to the best of our knowledge.

Response: To the best of our knowledge, there are no long-term ecological studies on pathogenic V.parahaemolyticus and V.alginolyticus. This is the main obstacle behind lack of conclusive outcomes of such investigations. According to Chen et al., 2017, the pathogenicity mechanism of V. alginolyticus is not yet fully understood. The pathogenicity of V. parahaemolyticus and V. alginolyticus is the result of a combination of multiple virulence factors. Studies on the mechanisms by which the environmental signals integrate into the genetic circuit to regulate the expression of these virulence factors would provide novel strategies to prevent and treat infection caused by vibrios.( LingzhiLi et al.,2019). Moreover isolation and culturing of pathogenic vibrios still remains as a challenge.
